# Analgesic Effect of a Novel Intravenous Ibuprofen-Low-Dose Tramadol Combination: A Multimodal Approach to Moderate-to-Severe Postoperative Dental Pain

**DOI:** 10.3390/pharmaceutics17101248

**Published:** 2025-09-24

**Authors:** M. Rosario Salas-Butrón, Leonor Laredo-Velasco, Ana B. Rivas-Paterna, Aránzazu González-Corchon, Mario F. Muñoz-Guerra, Alberto M. Borobia, Julio J. Acero-Sanz, Carla Pérez-Ingidua, Francisco Abad-Santos, Jose-Luis Cebrián, María Ángeles Gálvez-Múgica, Irene Serrano-García, Carmen Portolés-Díez, Lucia Llanos, Dolores Martínez, Nuria Sanz, Carlos Calandria, Emilio Vargas-Castrillón, Rafael Martín-Granizo, Antonio Portolés-Pérez

**Affiliations:** 1Servicio de Farmacología Clínica, Hospital Clínico San Carlos, IdISSC, 28040 Madrid, Spain; mariadelrosario.salas@salud.madrid.org (M.R.S.-B.); leonor.laredo@salud.madrid.org (L.L.-V.); carlap05@ucm.es (C.P.-I.); emilio.vargas@salud.madrid.org (E.V.-C.); 2Departamento de Farmacología y Toxicología, Facultad de Medicina, Universidad Complutense IdISSC, 28040 Madrid, Spain; 3Instituto de Investigación Sanitaria, Hospital Clínico San Carlos, IdISSC, 28040 Madrid, Spain; ab.rivas@enf.ucm.es (A.B.R.-P.); iserrag01@gmail.com (I.S.-G.); rafael.martingranizo@salud.madrid.org (R.M.-G.); 4Facultad de Enfermería, Universidad Complutense de Madrid, 28040 Madrid, Spain; 5Servicio de Cirugía Oral y Maxilofacial, Hospital Clínico San Carlos, IdISSC, 28040 Madrid, Spain; 6Servicio de Cirugía Oral y Maxilofacial, Hospital La Princesa, 28006 Madrid, Spain; 7Facultad de Medicina, Universidad Autónoma de Madrid, 28029 Madrid, Spainfrancisco.abad@salud.madrid.org (F.A.-S.); 8Servicio de Farmacología Clínica, Hospital La Paz, IDIPAZ, 28046 Madrid, Spain; 9Servicio de Cirugía Oral y Maxilofacial, Hospital Ramón y Cajal, 20034 Madrid, Spain; 10Servicio de Farmacología Clínica, Hospital La Princesa, IIS-Princesa, 28006 Madrid, Spain; 11Servicio de Cirugía Oral y Maxilofacial, Hospital la Paz, 28046 Madrid, Spain; 12Research Pharmacology Unit, Hospital Ramón y Cajal, IRyCIS, 20034 Madrid, Spain; 13Servicio de Anestesiología, Reanimación y Tratamiento del Dolor, Hospital Clínico San Carlos, 28040 Madrid, Spain; mariadelcarmen.portoles@salud.madrid.org; 14Unidad de Investigación Clínica, Hospital Universitario Fundación Jimenez Díaz, IIS-FJD, 28040 Madrid, Spain; 15Servicio de Cirugía Oral y Maxilofacial, Fundación Jimenez Díaz, 28040 Madrid, Spain; 16Farmalider SA, 28108 Madrid, Spain; nuriasanz@farmalider.com (N.S.); carloscalandria@farmalider.com (C.C.)

**Keywords:** fixed-dose combinations, dental surgery, pain, analgesic drugs, opioid, NSAID, synergy

## Abstract

**Background:** Drug combinations with complementary mechanisms of action are able to achieve effective analgesia at lower doses, thereby reducing the risk of adverse effects (AEs). This study evaluated the analgesic efficacy and tolerability of two fixed-dose combinations (FDCs) of ibuprofen/tramadol (IBU/TRA) compared with tramadol and a placebo. **Methods:** This multicenter, randomized, double-blind, dose-finding, pilot clinical trial compared IBU/TRA (400/37.5 mg and 400/75 mg) with 100 mg of tramadol and a placebo in patients with moderate-to-severe pain following dental surgery. The primary endpoints were pain intensity at 6 h (PI_6h_) and the pain intensity difference from baseline to 6 h (PID_6h_). PID_7h_, the sum of pain intensity differences from baseline to 7 h (SPID_0–7h_), pain relief (PAR_7h_), total pain relief (TOTPAR_7h_), the use of rescue medication and AEs were also assessed. **Results:** Seventy-two patients were randomized and evaluated. Both FDCs showed superiority over the placebo for PI_6h_ and PID_6h_ (*p* < 0.05) but were not significantly different from 100 mg of tramadol. The statistical superiority of FDCs over the placebo was observed for PID_7h_, SPID_0–7h_, PAR_7h_ and TOTPAR_7h_. The percentage of patients receiving rescue medication was higher in the placebo (94.1%) and tramadol (52.6%) groups than the FDC groups (35.3% and 36.8% for 400/37.5 mg and 400/75 mg, respectively). A post hoc analysis showed that the FDCs had a superior analgesic efficacy to 100 mg of tramadol in the SPID_0–4h_ (*p* < 0.005). The incidence of AEs was comparable between treatment groups. **Conclusions:** Both FDCs of IBU/TRA provided superior analgesic efficacy compared to the placebo. We propose using SPID_0–4h_ as the preferred variable for evaluating the efficacy of this type of drug combination.

## 1. Introduction

Acute pain is common in hospitalized patients, whether due to an acute illness or surgical procedures. Most patients undergoing surgery report acute postoperative pain, but evidence suggests that less than half receive adequate postoperative pain relief [1]. Postoperative pain is often associated with increased healthcare costs, poor patient satisfaction, and a higher risk of developing chronic pain syndromes [2,3].

Pain is transmitted through multiple neural pathways, with each involving distinct mechanisms [4]. Due to this complexity, combining different classes of analgesics that each target specific pain mechanisms can enhance pain control. This approach, known as multimodal analgesia, improves pain management while reducing the required dosage of monotherapy drugs and consequently minimizing side effects [5,6].

Multimodal analgesia offers several advantages in the postoperative setting. First and foremost, the combination of agents with different analgesic mechanisms can lead to synergistic effects, thereby increasing their overall efficacy. In addition, this synergism allows individual drug doses to be reduced, minimizing dose-related adverse effects, particularly by reducing the need for opioids [7,8,9].

For many years, opioid analgesics have been used to manage pain and provided significant therapeutic benefits when administered appropriately [10,11,12]. Emerging evidence suggests that prescribing lower doses correlates with a reduced incidence of long-term opioid use and a decreased risk of developing opioid use disorder [13,14]. Nonsteroidal anti-inflammatory drugs (NSAIDs) serve as fundamental components of opioid-sparing strategies within multimodal pain management protocols. Several studies have shown that the combination of NSAIDs with low-dose tramadol significantly reduces pain scores and provides superior analgesic efficacy compared to analgesic drugs when used as monotherapy [15,16,17,18,19].

Ibuprofen is an NSAID, a derivate of propionic acid with analgesic, antipyretic and anti-inflammatory activity. Its mechanism of action is based on the inhibition of cyclooxygenase, resulting in decreased prostaglandin synthesis [20,21].

Tramadol, a centrally acting analgesic that is structurally related to codeine and morphine, consists of two enantiomers, both of which contribute to analgesic activity via different mechanisms. (+)-Tramadol and the metabolite (+)-O-desmethyl-tramadol (M1) act as agonists of the μ-opioid receptor. (+)-Tramadol inhibits serotonin reuptake, while (−)-tramadol inhibits norepinephrine reuptake, both contributing to the enhanced inhibitory modulation of pain transmission within the spinal cord [22,23]. Its pharmacological profile contributes to its efficacy and tolerability in the treatment of pain caused by various aetiologies (e.g., post-traumatic, biliary colic, obstetric or postoperative pain) [22,24,25].

The present pilot study was designed to evaluate the preliminary analgesic efficacy and tolerability of two fixed-dose combinations (FDCs) of intravenous ibuprofen arginate/tramadol HCl (IBU/TRA), each containing different low doses of tramadol, compared with 100 mg of tramadol HCl and a placebo for postoperative pain following the extraction of impacted third molars; this was in order to facilitate the selection of an optimal dose of tramadol for combination. In addition, given the complexity of the evaluation of the effects of painkillers with different mechanisms, doses and rates of action, this study aimed to test the method used (variables, measurement times, doses) for the design of future studies. This study was designed in compliance with the recommendations of the European Medicines Agency (EMA) [26].

## 2. Materials and Methods

This study was conducted at five centres in Spain (Hospital Clínico San Carlos, Hospital Universitario La Princesa, Hospital Universitario La Paz, Hospital Universitario Ramón y Cajal and Hospital Universitario Fundación Jiménez Díaz).

The protocol was authorized by the AEMPS (https://reec.aemps.es/reec/public/list.html (accessed on 20 June 2025), EudraCT Number: 2018-001412-30, see Appendix A) and approved by the Ethics Committee of the Hospital Clínico San Carlos (Internal Code: 18/243-R_M). The study was conducted in compliance with the ethical principles originating in or derived from the more recent Declaration of Helsinki, International Council for Harmonization Good Clinical Practice guidelines, and local regulatory requirements in force.

An informed consent form was signed by each subject prior to inclusion in the study, after receiving comprehensive information about the study design, objectives and risks.

### 2.1. Study Design

This was a multicentre, randomized, double-blind, dose-finding, parallel design, placebo and active-controlled, single-dose, pilot study that aimed to determine the appropriate dose of tramadol in combination with ibuprofen (IBU/TRA 400/37.5 mg or IBU/TRA 400/75 mg)—compared to a full dose of tramadol (100 mg) and the placebo when administered intravenously—for moderate-to-severe acute pain following the extraction of an impacted third molar tooth. This study also aimed to explore a more appropriate method for evaluating responses in this kind of study.

The study period was structured into three phases for each patient. First, a screening phase was conducted during the four weeks prior to randomization, which included pre-surgical procedures and concluded within four hours post-surgery, once the eligibility criteria had been confirmed. This was followed by the randomization and treatment administration visit, after which a seven-hour assessment period began; during this, patients recorded data using a diary. Finally, an end-of-study visit was carried out between five and nine days after randomization (7 ± 2 days), completing the clinical follow-up.

For randomization into the treatment groups, subjects rated their pain intensity (PI) using a Visual Analogue Scale (VAS) ranging from 0 to 100 mm. Subjects experiencing moderate-to-severe pain (VAS ≥ 55 mm) within 4 h after surgery were randomized to receive an intravenous single dose of any of the following study treatments:▪IBU/TRA 400/37.5 mg, intravenous, single dose;▪IBU/TRA 400/75 mg, intravenous, single dose;▪Tramadol HCl 100 mg, intravenous, single dose;▪Placebo, intravenous, single dose.

The drugs were infused via a pump over 30 min. All treatments were supplied by Farmalider S.A. (Madrid, Spain), and were administered under double-blinded conditions. The administration was standardized for all subjects according to a specific medication guide.

Subjects who did not experience adequate pain relief from the study medication were allowed to take rescue medication (RM), which consisted of 1 g paracetamol (if a second RM was required, 575 mg metamizole was administered). Subjects were allowed to take RM at any time, but were advised to wait at least 30 min after receiving the study medication to allow its effect to start.

### 2.2. Subjects

Healthy subjects ≥18 years of age scheduled to undergo the removal of ≥2 impacted third molars requiring bone remotion under local anesthesia, and who experienced moderate-to-severe pain after the extraction, were randomized to receive an intravenous single dose of the assigned study treatment. Other inclusion criteria included the ability to record necessary information on the diary, and the ability to comply with the study requirements.

The exclusion criteria included patients with a history of allergy or hypersensitivity to the study medication, rescue medication or any of its excipients; patients with a history of asthma; patients with a history of peptic ulceration, gastrointestinal disorders, gastrointestinal bleeding or other active bleeding; patients with a history of moderate-to-severe renal, hepatic or cardiac failure; patients with a history of active bleeding or coagulation disorders; patients with a history of epilepsy; patients with Crohn’s disease or ulcerative colitis; patients with a history of drug or alcohol dependence; patients with a history of any disease or disorder that, at the investigator’s discretion, could pose a risk to the patient or alter the results of the study; patients who had taken any analgesic or medicine that should not be administered due to the risk of interactions; patients who had received an experimental drug or used an experimental medical device within 30 days prior to the screening process; and pregnant or breastfeeding women. Additionally, patients were excluded if any complication of surgery that could interfere with the study’s procedures or assessments occurred.

Although the main objective of this pilot study was exploratory, a sample size of 72 patients was considered enough to assess the analgesic efficacy of FDCs compared with tramadol and the placebo, assuming a power of 80% and an overall significance level of 5% (two-sided). The expected difference in PI 6 h was assumed to be about 10 (SD 11) based on previous trials conducted by the sponsor.

A balanced per-centre computer-generated randomization list was prepared prior to the study, and sealed opaque envelopes were used to ensure that allocations were concealed. Randomization codes were given directly to the individuals responsible for dispensing the medication at the site and not otherwise involved in any part of the clinical trial.

### 2.3. Efficacy Assessment

The analgesic efficacy evaluation was based on patient assessments of PI using the VAS, ranging from 0 mm (“no pain”) to 100 mm (“worst imaginable pain”), and their assessments of pain relief (PAR), also using the VAS, ranging from 0 mm (“no relief”) to 100 mm (“complete relief”). Data were collected at the following specific time points: baseline and 0.25, 0.5, 0.75, 1, 1.25, 1.5, 1.75, 2, 2.5, 3, 3.5, 4, 4.5, 5, 5.5, 6, 6.5, and 7 h post administration. The percentage of responders and percentage of patients requiring RM were also evaluated. Patients who received RM remained at the study site for the full duration of the study.

### 2.4. Primary and Secondary Variables

The primary efficacy variables were PI at 6 h (PI_6h_) and pain intensity difference from baseline through 6 h post treatment (PID_6h_). Secondary efficacy variables included PID_7h_, the sum of pain intensity, differences from baseline to 7 h (SPID_0–7h_), PAR_7h_, total pain relief over 7 h (TOTPAR_7h_), the percentage of responders in terms of PI reduction (subjects who achieved at least a 20% PI reduction vs. baseline at 6 h) and the percentage of patients using RM. The inclusion of these secondary variables enhances the interpretation and consistence of the observed effects.

### 2.5. Safety

AEs were documented throughout the study based on spontaneous patient reports and indirect questioning. Each AE was classified by the investigator according to its relationship with the study medication (unrelated, unlikely, possible, probable, or definite), its intensity (mild, moderate, or severe) and its seriousness.

Safety assessments were conducted by evaluating clinically relevant changes from baseline following drug administration. These evaluations included physical examinations, vital signs (body temperature, blood pressure and heart rate) and laboratory safety parameters such as hematology, biochemistry and coagulation profiles.

### 2.6. Population and Statistics

The main analysis was performed in the intention-to-treat (ITT) population, which included all the randomized patients. The analysis was also carried out in the per-protocol (PP) population, which included all patients from the ITT population who did not experience any relevant protocol deviation with regard to the efficacy endpoints of the primary objective.

The primary efficacy variables, PI_6h_ and PID_6h_, were used to assess the analgesic efficacy and safety of the FDCs in comparison to tramadol and the placebo using the analysis of covariance (ANCOVA). The ANCOVA included the baseline VAS score to control for initial differences in pain levels between the groups. In addition, the intervention groups were compared with the control group to determine the efficacy of the treatment while accounting for baseline variability.

The secondary efficacy variables PID_7h_, PAR_7h_, SPID_0–7h_ and TOTPAR_7h_ were analyzed descriptively and supported the results of the primary efficacy endpoint. The percentage of patients achieving a PI reduction of at least 20% and the percentage of patients who required RM were tested using a chi-squared test. A descriptive analysis was conducted for the variable SPID_0–4h_ to explore the effect of the treatment at an intermediate time point, using a parameter that may be more sensitive to variations in the onset of drug action.

The last observation carried forward (LOCF) method was used for the imputation of missing data, including all observations made after patients received rescue medication. This procedure was applied to all the efficacy outcomes.

All statistical analyses were performed using R (version 3.5.2). A *p*-value less than 0.05 was considered statistically significant.

Safety variables were analyzed using descriptive statistics and were run on the safety population; in addition, the chi-2 was used to test the difference in frequencies. AEs were coded using the Medical Dictionary for Regulatory Activities (MedDRA) dictionary.

## 3. Results

### 3.1. Study Participants

A total of 72 patients (48 female and 24 male) were randomized and received the study treatment (Figure 1). The main efficacy analysis was run on the ITT population, which included 72 patients. The PP population comprised a total of 66 randomized patients, as six participants were excluded due to major protocol deviations related to efficacy endpoints of primary interest (±5% variation in dose received or discontinuation of the infusion of the study drug due to an adverse event).

In accordance with the exclusion criteria, no participants reported any relevant comorbidities or concomitant medications that could interfere with their perception of pain.

The demographic and baseline characteristics of different treatment groups were balanced between groups (Table 1). The mean (SD) age was 24.7 (5.29) years (range 18–44). The majority of patients were white (91.7%). The percentage of women was 66.7%. The baseline PI values per group are shown in Table 1. A total of 33 (45.8%) patients completed the study without the use of rescue analgesia.

### 3.2. Efficacy Results

#### 3.2.1. Primary Variables

The evolution of PI over time is shown in Figure 2. The PI_6h_ and PID_6h_ values in the IBU/TRA 400/37.5 mg and the IBU/TRA 400/75 mg groups were statistically different from those of the placebo group in the ITT population (see Table 2), showing a lower PI_6h_ and higher PID_6h_ for both FDCs vs. placebo. The analysis of the PP population showed the same results. However, no statistically significant differences were observed between the two FDCs and 100 mg tramadol.

#### 3.2.2. Secondary Variables

As shown in Table 2, the analysis of the secondary variables PID_7h_, SPID_0–7h_, PAR_7h_, and TOTPAR_0–7h_, evaluated at 7 h, corroborated the superior effect of both FDCs in comparison to the placebo after a dosing interval of 6 h. No significant differences were found for any secondary variable when comparing the FDCs vs. tramadol.

A total of 39 (54.2%) patients received RM (see Table 3). Regarding the percentage of responders at 7 h, the most effective results were observed in patients treated with the FDCs IBU/TRA: 400/37.5 mg (64.7%), 400/75 mg (68.4%). The tramadol group showed a 57.9% response rate, while the placebo group had a 17.3% response rate (see Table 4).

A post hoc descriptive analysis of SPID_0–4h_ showed the superior analgesic efficacy of the FDCs IBU/TRA compared to tramadol (400/37.5 mg, *p* 0.034; 400/75 mg, *p* < 0.001) (see Table 4).

### 3.3. Safety

Safety analysis was performed on 72 randomized patients. Overall, 24 patients (33.3%) experienced one or more adverse events, one of which was considered serious but not related to the study medication. No clinically relevant differences were identified in the incidence of AEs between treatment groups. The most common AEs were dizziness (6.9%), nausea (6.9%), and vomiting (5.6%) (Table 5). No deaths occurred. There were no clinically relevant changes in the vital signs or physical examination vs. baseline. Overall, FDCs were safe and well tolerated.

## 4. Discussion

Pain is a complex phenomenon involving multiple pathophysiological mechanisms, including peripheral nociceptor activation, central sensitization, inflammation and descending pain modulation [27,28]. Because of this complexity, effective pain management requires a multimodal approach that targets the different pathways involved in the generation and maintenance of pain. The current paradigm of drug combination therapy permits the use of low doses of individual agents, thereby diminishing the likelihood of AEs while achieving the intended analgesic efficacy [15,29,30,31,32,33].

Multiple datasets and clinical evidence support the assertion that rapid analgesia is essential to preventing sensitization and neuronal plasticity, which follow all nociceptive signals and are responsible for the maintenance of pain. Specifically, it has been demonstrated that administering effective and fast analgesia during the acute phase can prevent the development of permanent hyperalgesia and therefore reduce the risk of pain chronification [34].

Opioids are potent analgesics and are available in a wide variety of formulations, including IV, oral, and transdermal, which is useful in the postoperative setting. The current crisis of opioid abuse and misuse underscores the imperative for rigorous risk mitigation and precise dose optimization [35]. Opioid monotherapy remains a widely used and effective method for post-operative analgesia. However, monotherapy with opioids has various idiosyncratic or dose-limiting side effects that curtail their practical efficacy, and also exposes patients to serious adverse events.

Several studies have supported the use of analgesic combinations, such as diclofenac/tramadol, dexketoprofen/tramadol, and acetaminophen/tramadol, as an effective strategy for managing acute postsurgical pain, particularly within the framework of multimodal analgesia [2,10,31,36,37,38,39,40]. Clinical guidelines likewise endorse the rational use of drug combinations to optimize analgesia and minimize opioid consumption [13]. In the specific context of postoperative dental pain management, particularly following procedures such as third molar extractions, the combination of NSAIDs and opioids has proven to be an effective strategy [18,41,42]. This multimodal approach not only enhances analgesic efficacy but also enables the use of lower opioid dosages, thereby reducing the likelihood of adverse effects.

The synergy between ibuprofen and opiate drugs is due to their complementary mechanisms of action: NSAIDs act peripherally, and opioids act centrally. This analgesic efficacy may be greater than expected when their individual effects are added, enhancing the efficacy while enabling the required doses of each drug to be reduced.

Although this study focused on dental extraction, this analgesic strategy could be applicable to other acute pain conditions. Nevertheless, further research is needed to confirm its efficacy in different surgical contexts.

After an extensive review, this is the first published study to evaluate the analgesic effect of a combination of ibuprofen with low doses of tramadol administered intravenously. This pilot study provides essential insights into the efficacy and safety of a novel drug combination, thereby supporting the design of future clinical trials. Furthermore, the availability of intravenous formulations provides a valuable alternative when the oral route is not feasible.

The study was designed in accordance with the EMA guidelines on the clinical development of medicinal products for pain treatment and fixed-dose drug combinations, providing methodological consistency [26,43]. The study was a randomized and double-blinded trial, minimizing potential sources of experimental bias.

The drug doses tested were based on a prior pharmacodynamics study that employed oral administration and one pharmacokinetics study [44] that indicated that intravenous tramadol doses should not be reduced compared to oral ones; this study also noted the expected synergy of ibuprofen and tramadol.

This pilot study demonstrates that the tested FDCs (IBU/TRA 400/37.5 mg and 400/75 mg) provided better analgesia than the placebo at 6 h in patients with moderate-to-severe acute dental pain after a single dose. No statistically significant differences were observed between the FDCs and tramadol for either the main or secondary variables. However, given the small sample size, the analysis lacked sufficient statistical power to conclusively rule out potential differences. Given the potential relevance of the different pharmacokinetic profiles of the drugs involved in the study (i.e., the expected Tmax for Ibuprofen is 0.5 h, while it is 1 h for tramadol and 2 to 3 h for its main active metabolite [44]), a post hoc analysis of SPID_0–4h_ was conducted. This analysis revealed the significantly greater efficacy of both FDCs compared to tramadol monotherapy, suggesting that SPID may serve as a more suitable endpoint for comparing drugs with different mechanisms of action, onset times, and effect durations. The LOCF method was used to handle missing data, including all the pain evaluations after patients received rescue analgesia. Since tramadol reaches peak plasma concentrations approximately 2 h after a single dose in healthy adults, and more than 50% of patients in the tramadol group required rescue analgesia before this time, many of them were recorded as having no change in pain intensity. This may have underestimated the efficacy of tramadol. In contrast, FDCs IBU/TRA were significantly superior to placebo across all endpoints, despite the use of LOCF, in this case rescue medication use was significantly less frequent in the FDCs groups.

Although no statistically significant differences (*p* = 0.491) were observed between the two FDCs, the frequency of adverse events was lower with the 400/37.5 mg dose, suggesting a more favourable safety profile. Given that the difference in the analgesic effect between both FDCs is not considered clinically significant, this finding could support the use of lower tramadol doses in combination therapy to minimize the risk of AEs. Currently, there are no commercially available intravenous formulations combining ibuprofen with tramadol doses below 50 mg, and this combination is protected by patent [45]. Given the dose-dependent nature of tramadol-related AEs and the widespread use of opioid analgesics, further research is needed to evaluate the efficacy and safety of more finely tuned intravenous combinations of such drugs.

The main limitation of this study is its single-dose design, with its evaluation of the short-term analgesic efficacy of FDCs IBU/TRA compared to the placebo and tramadol. Additionally, the sample size and duration of the study may have limited the detection of rare or infrequent adverse events. Additionally, the use of result imputation by LOCF in patients who received RM represents an inherent limitation in the assessment of analgesia for a slower action drug; this could be partly mitigated through the evaluation of secondary variables.

## 5. Conclusions

The results of this study show that both IBU/TRA combinations are effective and safe compared to the placebo in patients with moderate-to-severe pain following dental surgery.

This study provided estimates of both the effect size and variability, as well as the selected efficacy parameters, confirming the consistent and sustained analgesic efficacy of the IBU/TRA combinations throughout the observation period. Therefore, fixed-dose combinations of ibuprofen and tramadol may provide a good opportunity to achieve effective analgesia at lower opiate doses.

We also suggest using the variable SPID for the evaluation of the analgesic effects of these kinds of drugs and their combinations.

## 6. Patents

Portolés A, Santé L, Salas M, Vargas E, Calandria C et al., inventors; Farmalider S.A. assignee. Combination of ibuprofen and tramadol for relieving pain, WO/2021/005129, 14 January 2021 [45].

## Figures and Tables

**Figure 1 pharmaceutics-17-01248-f001:**
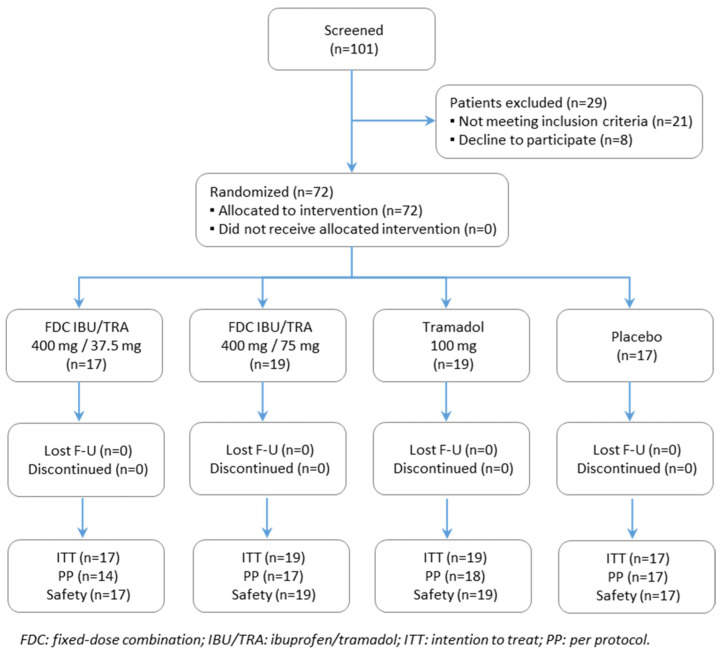
Participant flow chart of study. The ITT (intention-to-treat) population consisted of all randomized patients; safety population of all patients who received study drug; PP (per-protocol) population of all patients of the ITT who did not experience relevant protocol deviations related to efficacy endpoints of primary interest.

**Figure 2 pharmaceutics-17-01248-f002:**
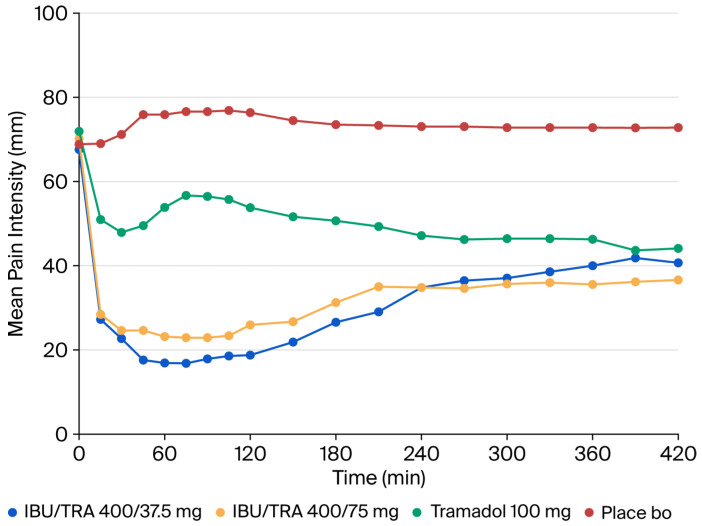
ITT population. Primary efficacy variable: pain intensity. IBU/TRA37.5: ibuprofen 400 mg/tramadol 37.5 mg; IBU/TRA75: ibuprofen 400 mg/tramadol 75 mg; Tramadol: Tramadol 100 mg.

**Table 1 pharmaceutics-17-01248-t001:** Patient demographic and baseline characteristics by treatment group (ITT population).

Demographic and Baseline Characteristics		IBU/TRA 400 mg/37.5 mg (n = 17)	IBU/TRA 400 mg/75 mg (n = 19)	Tramadol 100 mg (n = 19)	Placebo (n = 17)
Gender					
Female	n (%)	11 (64.7)	13 (68.4)	14 (73.9)	10 (58.8)
Male	n (%)	6 (35.3)	6 (31.6)	5 (26.3)	7 (41.2)
Age (years)					
Mean (SD)		25.5 (5.41)	24.9 (5.50)	23.8 (3.69)	25.2 (6.65)
Range		18–39	18–41	18–31	18–44
Body mass index					
Mean (SD)		24.7 (4.95)	23.4 (3.88)	22.6 (2.28)	23.6 (3.81)
Range		18.8–36.0	19.7–31.9	19.0–27.9	18.1–32.0
Pain intensity—baseline					
Mean (SD)		68.1 (12.88)	70.1 (9.95)	71.9 (10.38)	68.7 (12.80)
VAS ≤ 60	n (%)	6 (35.3)	4 (21.1)	3 (15.8)	6 (35.3)
VAS > 60	n (%)	11 (64.7)	15 (78.9)	16 (84.2)	11 (64.7)

ITT: intention to treat; VAS: Visual Analogue Scale; IBU/TRA: ibuprofen/tramadol.

**Table 2 pharmaceutics-17-01248-t002:** Summary of efficacy endpoints (ITT population).

Parameter	IBU/TRA 400/37.5 mg (n = 17)	IBU/TRA 400/75 mg (n = 19)	Tramadol 100 mg (n = 19)	Placebo (n = 17)	*p*-Value
PI_6h_ Mean (SD)	40.7 (26.69)	35.6 (31.96)	46.3 (34.14)	72.8 (25.30)	#P: 0.014 ##P: 0.003 #T: 0.941 ##T: 0.696
PID_6h_ Mean (SD)	27.5 (29.65)	34.5 (31.33)	25.6 (39.07)	−4.1 (25.69)	#P: 0.027 ##P: 0.003 #T: 0.998 ##T: 0.829
PID_7h_ Mean (SD)	27.2 (26.76)	33.4 (31.61)	27.8 (40.95)	−4.1 (25.69)	#P: 0.030 ##P: 0.005 #T: 0.999 ##T: 0.951
SPID_7h_ Mean (SD)	260.9 (165.66)	271.4 (188.83)	156.3 (245.08)	−33.9 (148.15)	#P: <0.001 ##P: <0.001 #T: 0.369 ##T: 0.261
PAR_7h_ Mean (SD)	55.1 (32.93)	57.1 (35.39)	45.3 (42.01)	10.9 (25.25)	#P: 0.002 ##P: 0.001 #T: 0.830 ##T: 0.041
TOTPAR_7h_ Mean (SD)	412.9 (203.47)	400.5 (234.35)	239.6 (256.87)	51.9 (138.86)	#P: <0.001 ##P: <0.001 #T: 0.084 ##T: 0.107

#P: IBU/TRA 400/37.5 mg vs. placebo; ##P: IBU/TRA 400/75 mg vs. placebo; #T: IBU/TRA 400/37.5 mg vs. Tramadol; ##T: IBU/TRA 400/75 mg vs. Tramadol.

**Table 3 pharmaceutics-17-01248-t003:** Use of rescue medication and patient responders within the evaluation period (7 h) after drug intake.

ENDPOINT	IBU/TRA 400/37.5 mg n = 17	IBU/TRA 400/75 mg n = 19	Tramadol 100 mg n = 19	Placebo n = 17	Total n = 72
Parameter	N (%)	N (%)	N (%)	N (%)	N (%)
Use of rescue medication	6 (35.3%)	7 (36.8%)	10 (52.6%)	16 (94.1%)	39 (54.2%)
Responder at 7 h	11 (64.7%)	13 (68.4%)	11 (57.9%)	3 (17.6%)	38 (52.8%)

**Table 4 pharmaceutics-17-01248-t004:** Post hoc SPID_0–4h_ analysis (ITT and PP populations).

Parameter	Population	IBU/TRA 400/37.5 mg n = 17	IBU/TRA 400/75 mg n = 19	Tramadol 100 mg n = 19	Placebo n = 17	*p*-Value
SPID_0–4h_ _Mean (SD)_	ITT	173.4 (87.18)	168.4 (103.89)	78.0 (132.24)	−21.5 (72.17)	#P: <0.001 ##P: <0.001 #T: 0.034 ##T: 0.041
	PP	176.5 (79.31)	170.6 (96.43)	79.1 (135.98)	−21.5 (72.17)	#P: <0.001 ##P: <0.001 #T: 0.042 ##T: 0.045

ITT: intention to treat; PP: per protocol; SPID: sum of pain intensity difference. #P: IBU/TRA 400/37.5 mg vs. placebo; ##P: IBU/TRA 400/75 mg vs. placebo; #T: IBU/TRA 400/37.5 mg vs. tramadol; ##T: IBU/TRA 400/75 mg vs. tramadol.

**Table 5 pharmaceutics-17-01248-t005:** Summary of adverse events by MedDRA preferred term (PT).

Adverse Event	IBU/TRA 400/37.5 mg n = 17	IBU/TRA 400/75 mg n = 19	Tramadol 100 mg n = 19	Placebo n = 17	Overall n = 72
	e/n (%)	e/n (%)	e/n (%)	e/n (%)	e/n (%)
Overall	7/4 (23.5)	15/8 (42.1)	14/8 (42.1)	5/4 (23.5)	41/24 (33.3)
Dizziness	1/1 (5.9)	3/2 (10.5)	1/1 (5.3)	1/1 (5.9)	6/5 (6.9)
Nausea	0/0 (0.0)	1/1 (5.3)	4/4 (21.1)	0/0 (0.0)	5/5 (6.9)
Vomiting	0/0 (0.0)	3/3 (15.8)	1/1 (5.3)	0/0 (0.0)	4/4 (5.6)
Vasovagal syncope	0/0 (0.0)	1/1 (5.3)	1/1 (5.3)	1/1 (5.9)	3/3 (4.2)
Headache	1/1 (5.9)	1/1 (5.3)	1/1 (5.3)	0/0 (0.0)	3/3 (4.2)
Hypotension	1/1 (5.9)	2/1 (5.3)	0/0 (0.0)	0/0 (0.0)	3/2 (2.8)
Stomach-ache	0/0 (0.0)	1/1 (5.3)	0/0 (0.0)	0/0 (0.0)	1/1 (1.4)
Diarrhea	0/0 (0.0)	0/0 (0.0)	1/1 (5.3)	0/0 (0.0)	1/1 (1.4)
Presyncope	0/0 (0.0)	1/1 (5.3)	0/0 (0.0)	0/0 (0.0)	1/1 (1.4)
Alanine Aminotransferase increased	1/1 (5.9)	0/0 (0.0)	0/0 (0.0)	0/0 (0.0)	1/1 (1.4)
Aspartate Aminotransferase increased	1/1 (5.9)	0/0 (0.0)	0/0 (0.0)	0/0 (0.0)	1/1 (1.4)
Gamma glutamyltransferase increased	1/1 (5.9)	0/0 (0.0)	0/0 (0.0)	0/0 (0.0)	1/1 (1.4)
Dysesthesia of upper extremity	0/0 (0.0)	0/0 (0.0)	0/0 (0.0)	1/1 (5.9)	1/1 (1.4)
Hypoesthesia teeth	0/0 (0.0)	0/0 (0.0)	1/1 (5.3)	0/0 (0.0)	1/1 (1.4)
Cough	0/0 (0.0)	0/0 (0.0)	0/0 (0.0)	1/1 (5.9)	1/1 (1.4)
Flu-like symptoms	0/0 (0.0)	1/1 (5.3)	0/0 (0.0)	0/0 (0.0)	1/1 (1.4)
Cold	1/1 (5.9)	0/0 (0.0)	0/0 (0.0)	0/0 (0.0)	1/1 (1.4)
Tonsillitis	0/0 (0.0)	0/0 (0.0)	0/0 (0.0)	1/1 (5.9)	1/1 (1.4)
Abscess oral	0/0 (0.0)	0/0 (0.0)	1/1 (5.3)	0/0 (0.0)	1/1 (1.4)
Jaw pain	0/0 (0.0)	1/1 (5.3)	0/0 (0.0)	0/0 (0.0)	1/1 (1.4)
Dysmenorrhoea	0/0 (0.0)	0/0 (0.0)	1/1 (5.3)	0/0 (0.0)	1/1 (1.4)
Alveolitis of jaw	0/0 (0.0)	0/0 (0.0)	1/1 (5.3)	0/0 (0.0)	1/1 (1.4)
Genitourinary infection	0/0 (0.0)	0/0 (0.0)	1/1 (5.3)	0/0 (0.0)	1/1 (1.4)

e: events; n (%): number and % of patients showing AEs.

## Data Availability

The data are available on reasonable request from the corresponding author due to confidentiality reasons.

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
