# Peer review of "Analgesic Effect of a Novel Intravenous Ibuprofen-Low-Dose Tramadol Combination: A Multimodal Approach to Moderate-to-Severe Postoperative Dental Pain"

_pharmaceutics, 2025, doi:10.3390/pharmaceutics17101248_

Round 1

Reviewer 1 Report

Comments and Suggestions for Authors

Dear Authors,

 I have read the manuscript and I send you my comments:

1) Line 97-100: Please add (the) reference (s)

2) Line 107-108: please explain

3) Methods: please add the  power calculation

4) Results: please add comorbidity and drug used, to evaluate the appropriate treatment with tramadol in these patients

Author Response

Question 1:

  1. Line 97-100: Please add (the) reference (s)

Response:

Thank you for your suggestion. We have added the appropriate references to support the statement:

“13. Dowell D, Ragan KR, Jones CM, Baldwin GT, Chou R. CDC Clinical Practice Guideline for Prescribing Opioids for Pain - United States, 2022. MMWR Recomm Rep. 2022 Nov 4;71(3):1-95. Doi: 10.15585/mmwr.rr7103a1”

“14, Decker H, Wu CL, Wick E. Multimodal Pain Control in Surgery 2020. Adv Surg. 2021 Sep;55:147-157.Doi: 10.1016/j.yasu.2021.05.011”

Question 2:

  1. Line 107-108: please explain

Response:

Thank you for your comment. We have revised the manuscript to provide a more detailed and clearer explanation of the analgesic activity of tramadol. The revised paragraph now reads as follows:

“… (+)-Tramadol and the metabolite (+)-O-desmethyl-tramadol (M1) act as agonists of the μ-opioid receptor. (+)-Tramadol inhibits serotonin reuptake, while (–)-tramadol inhibits norepinephrine reuptake, both contributing to enhanced inhibitory modulation of pain transmission within the spinal cord. Its pharmacological profile, contributes to the efficacy and tolerability in the treatment of pain of various aetiologies (e.g. post-traumatic, biliary colic, obstetric or postoperative pain) [22, 23]”

Question 3:

  1. Methods: please add the power calculation

Response:

Thank you for your comment. The power calculation was already included in the manuscript (line 190-191); however, we have modified the paragraph for clarity. The revised paragraph now reads as follows:

“Although the main objective of this pilot study was exploratory, a sample size of 72 patients was considered enough to assess the analgesic efficacy of FDCs compared with tramadol and placebo assuming a power of 80% and an overall significance level of 5% (two-sided). The expected difference in PI 6h is assumed to be about 10 (SD 11) based on previous trials conducted by the sponsor”.

Question 4:

  1. Results: please add comorbidity and drug used, to evaluate the appropriate treatment with tramadol in these patients

Response:

Thank you for your comment. We appreciate the importance of considering comorbidities and drug us when evaluating the suitability of tramadol treatment. However, as this study was conducted in the context of an acute pain model, specifically dental extraction, it is not expected that patients presented with significant pre-existing pathologies. Furthermore, our exclusion criteria were designed to eliminate patients with relevant comorbidities or chronic medication use that could interfere with pain perception or tramadol metabolism.

We have added the following sentence to the Results section: “In accordance with the exclusion criteria, no participants reported any relevant comorbidities or concomitant medication that could interfere with pain perceptions”

Reviewer 2 Report

Comments and Suggestions for Authors

The research article “Analgesic Effect of a New Intravenous Combination of Ibuprofen and Low-Dose Tramadol: A Randomized, Double-Blind, Placebo and Active-Controlled Dose-finding Study in Patients with Postoperative Dental Pain” was thoroughly evaluated and reviewed. The manuscript reported the analgesic efficacy and tolerability of two fixed-dose combinations of ibuprofen /tramadol compared with tramadol and placebo in patients with moderate to severe pain following dental surgery.

Generally, the study appears well-organized and of great interest for the readers. However, some issues should be addressed prior to publication. 

  1. Title is very lengthy and looks awkward. I suggest revision of the title.
  2. Names and affiliations of the authors should be formatted as per journal guide lines.
  3. Abstract: The author claimed “Drug combinations with complementary mechanisms of action offer the potential to achieve effective analgesia at lower doses, thereby reducing the risk of adverse effects (AEs). This statement raises question if the dose is decreased from normal to low this may also drop their efficacy this statement should be justified.

from normal to low this may also drop their efficacy this statement should be justified.

  1. The term multimodal should explain in the light of current study.
  2. Line 111; the spell of “aetiologies “should be checked.
  3. Line 158; spaces between words should be reduced.
  4. The author should justify weather this type of analgesic effect may also observe in the other than dental surgeries or it may be only applicable for dental/tooth extraction post-operative pain?
  5. Cite more recent studies from the last 5 years to highlight novelty.
  6. How the primary variables can be co-related to secondary variable, give brief description.
  7. The sample size for the study is too small, I recommend increasing it up to at least 200 individuals.
  8. There is a huge gender variation among participants 48 female: 24 male why it so?
  9. As this study is restricted to dental post-operative pain so the discussion section should be not generalized to pain rather it should stick to dental pain.
  10. The author should provide more references to discuss the current study in the light of updated literature
  11. The study needs to be compared with already available combination therapies for the treatment of dental pain.
  12. The authors designed this research study for a single dose. Further studies are advised to optimize the dose.
  13. The author should re-write the conclusion section in the light of key aspects of results.
  14. The author should provide more references up to 45 at least adding it to introduction and discussion section to strengthen the author claim.
  15. It would be better if the author provides patient consent form.
Comments on the Quality of English Language

Should be checked from a native speaker.

Author Response

Question 1:

  1. Title is very lengthy and looks awkward. I suggest revision of the title.

Response:

Thank you for your suggestion. We have revised the title, making it shorter. The new proposed title is: “Analgesic Effect of a Novel Intravenous Ibuprofen-Low-Dose Tramadol Combination: A Multimodal Approach to Moderate-to-Severe Postoperative Dental Pain”

Question 2:

  1. Names and affiliations of the authors should be formatted as per journal guidelines.

Response:

Thank you for your observation. The names and affiliations of the authors have been revised to comply with the journal’s formatting guidelines.

Question 3:

  1. Abstract: The author claimed “Drug combinations with complementary mechanisms of action offer the potential to achieve effective analgesia at lower doses, thereby reducing the risk of adverse effects (AEs). This statement raises question if the dose is decreased from normal to low this may also drop their efficacy this statement should be justified.

Response:

Thank you for your comment. We agree that reducing the dose of individual agents in a combination therapy could potentially compromise efficacy. However, the rationale behind combining drugs with complementary mechanisms of action, such as opioid and non-opioid analgesics, is precisely to achieve synergistic effects. This synergy allows for effective pain control at lower doses of each drug.

The purpose of our study is precisely to evaluate whether a lower dose of tramadol, when combined with ibuprofen, can maintain analgesic efficacy through complementary mechanisms of action. This approach is based on the hypothesis that synergistic interactions between agents with distinct mechanisms may allow for dose reduction without loss of therapeutic effect. In fact, this concept has led to the development of a patented formulation Patent, which supports the scientific and clinical relevance of our investigation.

The statement mentioned in the abstract is justified by the references provided in the introduction (see lines 90–94) “Multimodal analgesia offers several advantages in the postoperative setting. First and foremost, the combination of agents with different analgesic mechanisms can lead to synergistic effects, thereby increasing overall efficacy. In addition, this synergism allows individual drug doses to be reduced, minimizing dose-related adverse effects, particularly by reducing the need for opioids [7-9]

Question 4:

  1. The term multimodal should explain in the light of current study.

Response:

Thank you for your comment but the term multimodal is explained in the manuscript, specifically in the paragraph at lines 86–89 where its meaning is clarified in the context of pain management.

Due to this complexity, combining different classes of analgesics, each targeting specific pain mechanisms, can enhance pain control. This approach, known as multimodal analgesia...”

Question 5:

  1. Line 111; the spell of “aetiologies “should be checked.

Response:

Thank you for your comment. We have reviewed the term “aetiologies” and confirmed its correct usage according to British English spelling.

Question 6:

  1. Line 158; spaces between words should be reduced.

Response:

Thank you for your comment. The extra spaces in line 158 have been corrected.

Question 7:

  1. The author should justify weather this type of analgesic effect may also observe in the other than dental surgeries or it may be only applicable for dental/tooth extraction post-operative pain?

Response:

Thank you for your thoughtful comment. The current study was specifically designed to evaluate analgesic efficacy in the context of acute postoperative pain following dental extraction. This is a well-established and standardised model for assessing short-term pain interventions. While the findings are directly applicable to dental surgery, the pharmacological rationale, based on complementary mechanisms of action, suggests that similar analgesic effects could potentially be observed in other acute pain settings. However, we acknowledge that extrapolation to other surgical contexts should be approached with caution and that further studies are needed to confirm efficacy in other clinical scenarios.

We have added a statement to the Discussion section to clarify this point: “Although this study focused on dental extraction, this analgesic strategy could be applicable to other acute pain conditions. Nevertheless, further research is needed to confirm its efficacy in different surgical contexts”.

Question 8:

  1. Cite more recent studies from the last 5 years to highlight novelty.

Response:

Thank you for your valuable suggestion. We have revised the manuscript to include additional references. These updated citations are now incorporated in the Introduction and Discussion sections, where relevant.

Question 9:

  1. How the primary variables can be co-related to secondary variable, give brief description.

Response:

Thank you for your comment. The primary variable of our study quantifies the sensory experience of pain perceived by the patient. The secondary variables (i.e., SPID, pain relief and TOTPAR) are conceptually and clinically relevant to the study. The SPID0-7h allows for assessment of pain progression. Pain relief is an outcome directly influenced by the reduction in pain intensity and TOTPAR0-7h assesses total pain relief. The other study variables such as the use of rescue medication and the percentage of responders, are complementary measures of treatment efficacy. We have added a brief description of this relationship to the manuscript (see the 'Methods' section, subsection 2.4, 'Primary and secondary variables').

“The inclusion of these secondary variables enhances the interpretation and consistence of the observed effects.”

Question 10:

  1. The sample size for the study is too small, I recommend increasing it up to at least 200 individuals.

Response:

Thank you for your comment. We acknowledge the importance of an adequate sample size. However, this study was designed as a pilot study, and the sample size of 72 patients aligns with the range for exploratory research. It should be noted that, the goal was to evaluate the preliminary efficacy and feasibility of the method to design future studies.

Question 11:

  1. There is a huge gender variation among participants 48 female: 24 male why it so?

Response:

Thank you for your comment. Participants were included in the study according to predefined eligibility criteria, without stratification by gender. All eligible patients were randomized after inclusion. The observed gender imbalance reflects the actual distribution of patients who met the inclusion criteria during the recruitment period.

Question 12:

  1. As this study is restricted to dental post-operative pain so the discussion section should be not generalized to pain rather it should stick to dental pain.

Response:

Thank you for your comment. Dental pain has been used as a model due to its easy reproducibility which makes it applicable to other types of pain that share a similar mechanism.

As commented in response to question 7, we have added a paragraph in the Discussion section to clarify this point: “Although this study focused on dental extraction, this analgesic strategy could be applicable to other acute pain conditions. Nevertheless, further research is needed to confirm its efficacy in different surgical contexts”.

Question 13:

  1. The author should provide more references to discuss the current study in the light of updated literature

Response:

Thank you for your comment. New bibliographic citations have been added to the introduction and discussion sections of the manuscript.

Question 14:

  1. The study needs to be compared with already available combination therapies for the treatment of dental pain.

Response:

Thank you for your comment. A paragraph has been added to the Discussion section mentioning combined therapies with other drugs.

“Several studies have supported the use of analgesic combinations -such as diclofenac/tramadol, dexketoprofen/tramadol, acetaminophen/tramadol- as an effective strategy for managing acute postsurgical pain, particularly within the framework of multimodal analgesia [2,10,31,36-40]. Clinical guidelines likewise endorse the rational use of drug combinations to optimise analgesia and minimize opioid consumption [13]. In the specific context of postoperative dental pain management, particularly following procedures such as third molar extractions, the combination of NSAIDs and opioids has proven to be an effective strategy [18,41,42]. This multimodal approach not only enhances analgesic efficacy but also enables the use of lower opioid dosages, thereby reducing the likelihood of adverse effects..”

Question 15:

  1. The authors designed this research study for a single dose. Further studies are advised to optimize the dose.

Response:

Thank you for your comment. We agree that future studies should explore multiple dosing regimens in order to optimize the therapeutic effect. In that sense, this study aims to identify the effective dose as a preliminary step toward the development of multiple-dose studies, which are currently underway.

Question 16:

  1. The author should re-write the conclusion section in the light of key aspects of results.

Response:

Thank you for your suggestion. In accordance with your recommendation, the conclusion section has been revised to better reflect the key findings of the study.

“The results of this study show that both IBU/TRA combinations are effective and safe compared to placebo in patients with moderate to severe pain following dental surgery.

The study provided estimates of both the effect size and variability or selected efficacy parameters, confirming the consistent and sustained analgesic efficacy of the IBU/TRA combinations throughout the observation period.

Therefore, the fixed-dose combinations of ibuprofen and tramadol may provide a good opportunity to achieve effective analgesia at lower opiate doses.

The variable SPID is suggested as very appropriate for the evaluation of analgesic effects of these kinds of drugs and combinations.

Question 17:

  1. The author should provide more references up to 45 at least adding it to introduction and discussion section to strengthen the author claim.

Response:

Thank you for your comment. We have followed the editorial's recommendation to limit the number of references. However, in response to your comment, we have included additional references.

Question 18:

  1. It would be better if the author provides patient consent form.

Response:

Thank you for your comment. All participants provided written informed consent prior to inclusion in the study, as approved by the institutional ethics committee (this information is included in the manuscript, lines 130-132). While the full consent form is not included in the manuscript, it can be made available to the journal upon request.

Reviewer 3 Report

Comments and Suggestions for Authors

This pilot study examines the analgesic efficacy of two fixed-dose combinations (FDCs) of intravenous ibuprofen and tramadol compared to tramadol monotherapy and placebo in patients with moderate to severe pain following third molar extraction. Please revise the following errors.

Lines 95-98: "Emerging evidence suggests that prescribing lower doses correlates with a reduced incidence of long-term opioid use..." - This statement needs specific references to support the claim about dose-response relationships for long-term opioid dependence.

Lines 112-118: The study objectives are poorly defined. The phrase "exploring the best method for this kind of study" is too vague and needs specific methodological objectives.

Lines 135-141: Study design description lacks critical details about allocation concealment and sequence generation. The term "dose-finding" is misleading as only two doses were tested.

Lines 148-151: Pain intensity threshold (VAS≥55 mm) needs justification. Why not the more standard ≥40mm or ≥50mm used in other pain studies?

Lines 188-191: "Based on a previous studies conducted by the sponsor (confidential data)" is unacceptable for peer review. Complete sample size calculations with all assumptions must be provided.

Lines 350-353: "To the best of our knowledge, this is the first published study..." - This claim requires a systematic literature search to support. Current literature review appears inadequate.

Lines 360-362: The rationale for dose selection based on "prior pharmacokinetics study [28]" and "expected synergy" is insufficient. Quantitative dose-response modeling should have been performed.

Author Response

Question 1:

Lines 95-98: "Emerging evidence suggests that prescribing lower doses correlates with a reduced incidence of long-term opioid use..." - This statement needs specific references to support the claim about dose-response relationships for long-term opioid dependence.

Response:

Thank you for your comment. We have added specific references that support the statement.

“13. Dowell D, Ragan KR, Jones CM, Baldwin GT, Chou R. CDC Clinical Practice Guideline for Prescribing Opioids for Pain - United States, 2022. MMWR Recomm Rep. 2022 Nov 4;71(3):1-95. Doi: 10.15585/mmwr.rr7103a1.”

“14. Decker H, Wu CL, Wick E. Multimodal Pain Control in Surgery 2020. Adv Surg. 2021 Sep;55:147-157. Doi: 10.1016/j.yasu.2021.05.011.”

Question 2:

Lines 112-118: The study objectives are poorly defined. The phrase "exploring the best method for this kind of study" is too vague and needs specific methodological objectives.

Response:

Thank you for your comment. We have revised the manuscript to provide more detailed. The revised paragraph now reads as follows:

“The present pilot study was designed to evaluate the preliminary analgesic efficacy and tolerability of two fixed-dose combinations (FDCs) of intravenous ibuprofen arginate/tramadol HCl (IBU/TRA), each containing different low doses of tramadol, compared with tramadol HCl 100 mg and placebo in postoperative pain following extraction of impacted third molars with a view to the selection of an optimal dose of tramadol for combination. Also, given the complex evaluation of the effects of painkillers with different mechanisms, doses and rates of action, the study aimed to test the method (variables, measurement times, dosing) for the design of future studies.

Question 3:

Lines 135-141: Study design description lacks critical details about allocation concealment and sequence generation. The term "dose-finding" is misleading as only two doses were tested.

Response:

Thank you for your comment. We have added details regarding allocation concealment to the Methods section:

A balanced per centre computer-generated randomization list was prepared prior to the study and allocation concealment was ensured by sealed opaque envelopes. Randomization codes were provided directly to the individuals responsible for dispensing the study medication at the site who were not otherwise involved in any part of the clinical trial.”

Regarding the term dose-finding, only two dose levels were selected to minimize the complexity of the study. Furthermore, prior to this trial, a pharmacokinetic study using both intravenous and oral administration routes was conducted, which allowed for the selection of the most appropriate doses.

Reviewer 4 Report

Comments and Suggestions for Authors
  1. Sample Size Justification

Although it is stated that the sample size of 72 patients was sufficient based on prior internal data, this justification remains unclear. Include either the effect size or more detailed assumptions (e.g., expected difference in PI6h, standard deviation, dropout rate) to support the sample size.

  1. Primary Endpoint Selection & Interpretation

While PI6h and PID6h are commonly used, the authors propose SPID0–4h as a more relevant efficacy measure post hoc. This is a crucial claim.  Consider moving the SPID0–4h rationale from the discussion to the methods, as its post hoc nature may otherwise appear data-driven rather than hypothesis-driven.

  1. Comparative Efficacy vs. Tramadol

The lack of statistical difference between FDCs and tramadol in primary endpoints could be viewed as inconclusive rather than negative. Discuss whether the study was powered to detect superiority over tramadol, and interpret this non-difference more cautiously.

  1. Safety Interpretation of Higher Tramadol Dose (75 mg)

Authors conclude that 400/37.5 mg is preferable based on AE frequency. However, this dose was also less effective in SPID0–4h than 400/75 mg. A more balanced discussion is needed to weigh both the benefits (pain relief) and risks (side effects) of each dose. Also, please clarify if the differences in adverse events between groups were statistically significant.

  1. Bias from LOCF and Early RM Use

The LOCF imputation after rescue medication may introduce bias, especially if RM use occurred before the peak effect. Provide a sensitivity analysis excluding patients who received RM before 2 h to address this concern. Otherwise, acknowledge it more prominently in limitations.

  1. In Table 5, the format (e/n%) is unclear at first glance. Readers may misinterpret 'e' and 'n'.
    Add a footnote explaining "e: events, n: patients."
  2. 7-hour assessment post-dose is appropriate, but unclear why the study period is described as 5 weeks. Clarify what other visits occurred across this 5-week timeline post-dosing.
  3. Criteria around CNS-active medications (SSRIs, antipsychotics) are appropriate but not justified. Briefly explain the pharmacodynamic interaction risk, especially since tramadol has serotonergic activity.

Author Response

Question 1:

  1. Sample Size Justification

Although it is stated that the sample size of 72 patients was sufficient based on prior internal data, this justification remains unclear. Include either the effect size or more detailed assumptions (e.g., expected difference in PI6h, standard deviation, dropout rate) to support the sample size.

Response:

Thank you for your comment. We have modified the paragraph for clarity. The revised paragraph now reads as follows:

“Although the main objective of this pilot study was exploratory, a sample size of 72 patients was considered enough to assess the analgesic efficacy of FDCs compared with tramadol and placebo assuming a power of 80% and an overall significance level of 5% (two-sided). The expected difference in PI 6h is assumed to be about 10 (SD 11) based on previous trials conducted by the sponsor”.

Question 2:

  1. Primary Endpoint Selection & Interpretation

While PI6h and PID6h are commonly used, the authors propose SPID0–4h as a more relevant efficacy measure post hoc. This is a crucial claim.  Consider moving the SPID0–4h rationale from the discussion to the methods, as its post hoc nature may otherwise appear data-driven rather than hypothesis-driven.

Response:

Thanks for your comment. Although it is stated at the methods section as a post hoc descriptive analysis for the variable SPID0-4h in order to explore the effect at an intermediate time point, we agree with you that it is not only the time point but also the parameter what is relevant.

Accordingly, the text has been adapted to clarify this aspect:

“A post hoc descriptive analysis was conducted for the variable SPID0-4h to explore the treatment effect at an intermediate time point, using a parameter that may be more sensitive to variations in the onset of drug action.”

Question 3:

  1. Comparative Efficacy vs. Tramadol

The lack of statistical difference between FDCs and tramadol in primary endpoints could be viewed as inconclusive rather than negative. Discuss whether the study was powered to detect superiority over tramadol, and interpret this non-difference more cautiously.

Response:

Agree, the study was designed as exploratory, and capable to detect differences in case they were large enough.

The analysis of comparative effect to Tramadol has low statistical power, although given the distribution of results it is very unlikely that difference could exist.

Attending to your comment, the text has been modified to interpret these results more cautiously.

Attending to your comment, the text has been modified as follows:

“No statistically significant differences were observed between the FDCs and tramadol for either the main or secondary variables. However, given the small sample size, the analysis lacked sufficient statistical power to conclusively rule out potential differences”.

Question 4:

  1. Safety Interpretation of Higher Tramadol Dose (75 mg)

Authors conclude that 400/37.5 mg is preferable based on AE frequency. However, this dose was also less effective in SPID0–4h than 400/75 mg. A more balanced discussion is needed to weigh both the benefits (pain relief) and risks (side effects) of each dose. Also, please clarify if the differences in adverse events between groups were statistically significant.

Response:

Thanks for your comment. In terms of safety, the % of patients suffering AE was 23.5% vs 42.1% (lower vs upper dose), and per the SPID0-4h results the lower dose appears as preferable (173.4 vs 168.4). Upon these results we considered as preferable the combination with a lower dose of tramadol. Anyway, the differences found in safety or efficacy among these combinations were not statistically significant.

Attending to your comment, the text has been modified as follows:

“Although no statistically significant differences were observed between the two FDCs, the frequency of adverse events was lower with the 400/37.5 mg dose, suggesting a more favourable safety profile. Given that the difference of the analgesic effect between both FDCs is considered not clinically significant, this finding could support the use of lower tramadol doses in combination therapy to minimize the risk of AEs”

Question 5:

  1. Bias from LOCF and Early RM Use

The LOCF imputation after rescue medication may introduce bias, especially if RM use occurred before the peak effect. Provide a sensitivity analysis excluding patients who received RM before 2 h to address this concern. Otherwise, acknowledge it more prominently in limitations.

Response:

Thanks for your comment. The evaluation of analgesia requires the use of imputation as far as the use of rescue medication is ethically mandatory.

We agree that LOCF as any other method could affect the evaluation of the effect, but in fact the patient receiving rescue medication is under the influence of another analgesic treatment which modifies pain evaluation. Among other methods, BOCF presumably would offer similar results, and imputation by the mean of the population not affected by rescue medication was not applicable in this study (almost all the patients in the placebo group received RM).

Attending to your comment, the limitation has been stated as follows:

“Additionally, the use of result imputation by LOCF in patients who received RM represents an inherent limitation in the assessment of analgesia, which can be partly mitigated through the evaluation of secondary variables.”

Question 6:

  1. In Table 5, the format (e/n%) is unclear at first glance. Readers may misinterpret 'e' and 'n'. Add a footnote explaining "e: events, n: patients."

Response:

Thanks for your comment. The format (e/n) in Table 5 is explained in the footnote to the table.

Question 7:

  1. 7-hour assessment post-dose is appropriate, but unclear why the study period is described as 5 weeks. Clarify what other visits occurred across this 5-week timeline post-dosing.

Response:

Thanks for your comment. The 5-week study period includes not only the 7-hour post-dose assessment but also the screening period prior to dosing and a safety follow-up visit.

To avoid any confusion, the paragraph is amended: “The study period was structured into three phases for each patient. First, a screening phase was conducted during the four weeks prior to randomization, which included pre-surgical procedures and concluded within four hours post-surgery, once eligibility criteria were confirmed. This was followed by the randomization and treatment administration visit, after which a seven-hour assessment period began, during which patients recorded data using a diary. Finally, an end of study visit was carried out between five and nine days after randomization (7 ± 2 days), completing the clinical follow-up.”

Question 8:

  1. Criteria around CNS-active medications (SSRIs, antipsychotics) are appropriate but not justified. Briefly explain the pharmacodynamic interaction risk, especially since tramadol has serotonergic activity.

Response:

Thanks for your comment. According to the technical sheet of tramadol, the use of tramadol with serotonergic drugs such as selective serotonin reuptake inhibitors (SSRIs), serotonin and norepinephrine reuptake inhibitors (SNRIs), monoamine oxidase inhibitors (MAOIs), tricyclic antidepressants and mirtazapine can lead to serotonin syndrome, which can be fatal.

However, following the editor’s remarks concerning potential overlap with existing publications, the affected section have been rewritten as presented below:

“Patients with a history of allergy or hypersensitivity to the study medication, rescue medication or any of its excipients; history of asthma; history of peptic ulceration, gastrointestinal disorders, gastrointestinal bleeding or other active bleeding, history of moderate to severe renal, hepatic or cardiac failure; active bleeding or coagulation disorders; history of epilepsy; Crohn’s disease or ulcerative colitis; history of drug or alcohol dependence; history of any disease or disorder that, at the investigator's discretion, could pose a risk to the patient or alter the results of the study; patients who had taken any analgesic or medicine that should not be administered due to the risk of interactions; patients who have received an experimental drug or used an experimental medical device within 30 days prior to screening process; pregnant or breastfeeding women. Additionally, patients were excluded if any complication of surgery occurred which could interfere with study procedures or assessments.“

Round 2

Reviewer 1 Report

Comments and Suggestions for Authors

none

Author Response

Thanks for your comments

Reviewer 2 Report

Comments and Suggestions for Authors

No further  comments.

Author Response

Thanks for your comments. The manuscript has been fully revised by a professional service of native speaker, as well as the figures and tables to better communicate the results.

Reviewer 4 Report

Comments and Suggestions for Authors

Although authors have made substantial improvements in the revised version, some comments remain unaddressed, including SPID0-4 4h rationale, safety balance, and LOCF bias. These points need explanation.

1.Primary endpoint & SPID0–4h

SPID0–4h rationale is still described mainly in results/discussion sections as post hoc. Methods mention it only as a post hoc descriptive analysis. It was suggested to move the SPID0–4h rationale upfront, but it still remains presented as an exploratory post-hoc analysis.

  1. Safety interpretation (37.5 mg vs 75 mg)

The authors recommend 400/37.5 mg as the preferred dose due to its lower AE frequency and minimal clinically significant analgesic difference. However, no statistical testing of AE differences is provided; instead, descriptive frequencies and a balance between efficacy (where SPID0–4h was higher for 75 mg) and safety are discussed.

  1. LOCF bias / early RM use

LOCF limitations are acknowledged, but no sensitivity analysis excluding patients who used rescue meds before 2h was added. It was asked either for analysis or a clear acknowledgment; the authors gave only a general limitation note.

Author Response

Thanks for your thoughtful revision. The manuscript has been fully revised by a professional native speaker service, as well as the figures and tables to better communicate the research. Please find below the answer to your questions, what I hope will be fully solved.

Question 1: Primary endpoint & SPID0–4h: SPID0–4h rationale is still described mainly in results/discussion sections as post hoc. Methods mention it only as a post hoc descriptive analysis. It was suggested to move the SPID0–4h rationale upfront, but it still remains presented as an exploratory post-hoc analysis.

Response: Thank you for your comment. We have moved the SPID a bit earlier in methods section, to be joined to the rest of variables, as well as given some more importance.

“A descriptive analysis was conducted for the variable SPID0-4h to explore the treatment effect at an intermediate time point, using a parameter that may be more sensitive to variations in the onset of drug action.”

Question 2: Safety interpretation (37.5 mg vs 75 mg): The authors recommend 400/37.5 mg as the preferred dose due to its lower AE frequency and minimal clinically significant analgesic difference. However, no statistical testing of AE differences is provided; instead, descriptive frequencies and a balance between efficacy (where SPID0–4h was higher for 75 mg) and safety are discussed.

Response: Thanks for your comment. In terms of safety, the % of patients suffering AE was 23.5% vs 42.1% (lower vs upper dose), and per the SPID0-4h results the lower dose appears as preferable (173.4 vs 168.4). Upon these results we considered as preferable the combination with a lower dose of tramadol. Anyway, the differences found in safety or efficacy among these combinations were not statistically significant.

Attending to your comment, the manuscript has been modified as follows:

Methods: “Safety variables were analyzed using descriptive statistics and were run on the safety population, also chi-2 was used to test difference in frequencies…”

Discussion: “Although no statistically significant differences (p= 0.491) were observed between the two FDCs, the frequency of adverse events was lower with the 400/37.5 mg dose, suggesting a more favourable safety profile. Given that the difference of the analgesic effect between both FDCs is considered not clinically significant, this finding could support the use of lower tramadol doses in combination therapy to minimize the risk of AEs.”

Question 3: LOCF bias / early RM use: LOCF limitations are acknowledged, but no sensitivity analysis excluding patients who used rescue meds before 2h was added. It was asked either for analysis or a clear acknowledgment; the authors gave only a general limitation note

Response: Thanks for your comment. The evaluation of analgesia requires the use of imputation as far as the use of rescue medication is ethically mandatory.

We agree that LOCF, as any other method, could affect the evaluation of the effect, but in fact the patient receiving rescue medication is under the influence of another analgesic treatment which modifies pain evaluation. Among other methods, BOCF presumably would offer similar results, and imputation by the mean of the population not affected by rescue medication was not applicable in this study (almost all the patients in the placebo group received RM).

We agree that the use of result imputation by LOCF could underestimate the effect of a medication with slower action.

Attending to your comment, the limitation has been stated as follows:

“…Additionally, the use of result imputation by LOCF in patients who received RM represents an inherent limitation in the assessment of analgesia of a slower action drug, which can be partly mitigated through the evaluation of secondary variables.”